# Response of Spider and Epigaeic Beetle Assemblages to Overwinter Planting Regimes and Surrounding Landscape Compositions

**DOI:** 10.3390/insects14120951

**Published:** 2023-12-15

**Authors:** Hainan Chong, Yulin Zhu, Qian Lai, Song Wu, Ting Jiang, Dandan Zhang, Haijun Xiao

**Affiliations:** 1School of Grassland Science, Beijing Forestry University, Beijing 100083, China; chonghn15@bjfu.edu.cn; 2Institute of Entomology, Jiangxi Agricultural University, Nanchang 330045, China; zhuyl0821@163.com (Y.Z.); ggjopj@hotmail.com (Q.L.); w15179282987@hotmail.com (S.W.); jiangting0315@126.com (T.J.); 3Institute of Biological Resources, Jiangxi Academy of Sciences, Nanchang 330096, China

**Keywords:** landscape, overwinter rotation pattern, rice, oilseed rape, predators, diversity, functional traits

## Abstract

**Simple Summary:**

Predatory spider and epigaeic beetle assemblages are major natural enemies (NEs) in agricultural ecosystems. The effects of non-crop natural habitats in supporting predatory NEs are well known for summer crop systems; however, limited attention has been given to summer–winter crop rotation transaction systems and during overwintering periods, especially in small-scale farmland systems. Here, we examine the effect of winter planting regimes and landscape composition on the species diversity and functional diversity of ground predators (spiders and carabids) in eight gradient landscape sites in southern China over two consecutive winters (2019/2020 and 2020/2021). Our work reveal that the spiders were more abundant and had a higher activity density in fallow rice fields (FRs) than oilseed rape fields (OSRs); however, the ground beetles were more abundant in OSRs than that in FRs. The composition of spider assemblages was impacted by semi-natural habitats (SNHs) during overwintering, while ground beetle assemblages were influenced mainly by overwintering planting patterns. The results suggest that different planting regimes and SNHs are a strategic way to enhance these ground predators. To conserve and improve predator diversity during overwintering, we suggest maintaining a diversity of planting regimes and conserving local semi-natural habitats.

**Abstract:**

The rotation patterns of summer rice–winter oil seed rape and summer rice–winter fallow are the main planting regimes in the rice ecosystem in southern China. However, the impact of local rotation patterns and landscape factors on the overwintering conservation of predators in spider and epigaeic beetle assemblages remains poorly understood. Here, we investigate the diversity and density of spiders and beetles over two consecutive winters (2019/2020 and 2020/2021), focusing on the impact of two rotation patterns (rice–fallow and rice–oilseed rape) and surrounding landscape compositions on predator diversity. The main findings of our research were that spiders were more abundant and had a higher activity density in the fallow rice fields (FRs) compared to the oilseed rape fields (OSRs), whereas ground beetles exhibited the opposite pattern. Specifically, fallow rice fields supported small and ballooning spiders (e.g., dominant spider: *Ummeliata insecticeps*), while OSRs supported larger ground beetles (e.g., dominant beetles: *Agonum chalcomus* and *Pterostichus liodactylus*). Moreover, the composition of spider assemblages were impacted by semi-natural habitats (SNHs) during overwintering, while ground beetle assemblages were influenced by overwinter planting patterns. Overall, our results suggest that different planting regimes and preserving semi-natural habitats are a strategic way to enhance species diversity and functional diversity of ground predators. It is, therefore, recommended that to conserve and improve predator diversity during overwintering, land managers and farmers should aim to maintain diverse planting regimes and conserve local semi-natural habitats.

## 1. Introduction

The intensification and expansion of agricultural land has proven to be a serious threat to the biodiversity of the ecosystem [1]. Intensification and expansion of agriculture improve land use efficiency, but they also result in the simplification of polycultures into monocultures [2,3]. This leads to a decline in natural habitats and crop diversity, which may be beneficial for pest outbreaks [4,5,6]. Reasonable land use forms and rotation patterns at the large-scale landscape level, habitat patch types, crop tillage or cultivation patterns, variety mix, and layout on a field-by-field basis can mitigate this impact [7,8]. For example, planting wildflowers and leaving grassland and hedgerow strips beside cultivated lands can promote natural enemies’ diversity and improve the colonization of ground predators [9,10]. Additionally, crop rotation, cover crops, and variety combinations can also greatly reduce pest population density and the incidence of pathogen infection, thus improving crop health [11]. Although the effects of natural habitat systems in supporting natural enemies are well-established for summer crop systems, they have received little attention during overwintering periods and for summer–winter crop rotation systems.

Communities of spiders (Araneae) and ground beetles (Carabidae) are often studied in agricultural ecosystems, not only as ecological indicators but also for top-down effects in decreasing focal pests like Diptera and Hemiptera [12,13,14]. A functional diversity of Araneae and Carabidae species helps us to understand the potential of natural enemies in providing ecosystem services, and it is also an indicator of organisms that are sensitive to environmental changes [15,16]. For spiders, body size, dispersal ability, and feeding traits, which reflect the history of life mechanisms in resource use, and the adaptive capacity to cope with environmental changes, are well-known [17,18]. Carabid beetles are also sensitive to environmental changes. Their body size, trophic levels, and wing morphology are the main traits considered [19,20]; these traits explain many life cycle events and the mobility response to different habitats’ disturbance [21]. Many studies have been conducted on the functional characteristics of predatory natural enemies, and the results reveal that larger, more mobile species always respond to landscapes with higher diversity types [22]. However, simple habitat filtering can lead to the homogenization of species; therefore, it is essential to maintain land use diversity [23]. Before now, there were limited studies investigating the functional diversity of ground predators during overwintering for different planting regimes. However, improving our understanding of this aspect can provide additional insight into how best to design efficient management strategies that protect these predators’ diversity and improve ecosystem services.

In Europe, oilseed rape fields and adjacent agri-environmental schemes (AESs) have increased the proportion of spider and carabid species [24]. In subtropical China, yearly rotation pattern of summer rice–winter rape is one of the most common land use system. This planting pattern improves paddy soil fertility and increases arthropod diversity, which are beneficial to the sustainability of agroecosystems [25,26,27]. Alternatively, summer rice–winter fallow rotation is another planting method seen in southern China, and fallow rice fields are possibly an undervalued resource for natural enemies in an agroecosystem. In this case, although fallow rice fields do not have the same variety of resources as in the summer season, they are subject to fewer disturbances, which provides a suitable habitat for rice field species in agricultural landscapes [28,29]. Fallow fields are often a habitat for soil animals such as Acarina and Collembola, which can provide alternative resources to maintain natural enemies’ survival during the overwintering season [30,31].

Over the past decades, the cultivation regimes and farmland consolidation have experienced rapid change amid efforts to simplify and homogenize the agricultural landscape [32,33]. However, this raises questions about the conservation of farmland natural enemies’ biodiversity and their associated biocontrol services, for example, in relation to farmland management and planting patterns, for which there are large knowledge gaps.

To begin to fill these, we examined the effects of different planting regimes and landscape compositions on the species diversity and functional diversity of ground predators (spiders and carabids) at thirteen sampling sites in Jiangxi Province, China. Specifically, we explored (i) the population dynamics of ground carabid beetles and spiders in different planting regimes of oilseeds rape fields (OSRs) and fallow rice fields (FRs); (ii) the composition of ground carabid beetles and spiders in the different planting regimes; and (iii) for the overwinter planting regimes and surrounding landscape composition factors, which one most affected ground carabid beetle and spider assemblages, and whether this effect depended on the planting regimes.

## 2. Materials and Methods

### 2.1. Study Area

This study was conducted over two consecutive winter seasons (2019/2020 and 2020/2021). Each winter season, eight sampling sites (for each site, the mean area of selected focal field = 1100 m^2^, range: 720–1980 m^2^) were selected in northern Jiangxi Province, China (E 115°53′, N 28°41′) (Figure 1). The sampling sites were independently isolated with a minimum distance of 6.3 km. At each site, two adjacent fields of fallow rice field (FR) and oilseed rape field (OSR) were selected for pitfall trapping set up, which ensured the same effect of landscape on recolonization of ground predators [34,35]. In addition, all the selected focal oilseed rape fields were planted with the same traditional rape bred cultivar (Yang–Guang 2009) to avoid a potential cultivar influence on the arthropod community.

The surrounding landscape composition of focal fields at a scale of 1000 m radius was quantified using remote sensing image data. High-light map images were printed and confirmed the land use according to the ground truth during the sample period. Furthermore, ArcGIS 10.6 was used to digitize the landscape context. Using FRAGSTATS 4.2, the landscape compositions and diversity were calculated. The landscape composition includes the percentage area of oilseed rape fields, the percentage area of fallow rice fields, and the percentage area of semi-natural habitats, while Shannon diversity represents the landscape diversity (Appendix A).

### 2.2. Sampling 

Spiders and carabids in the field were sampled with pitfall traps. In the center of each FR and OSR field, nine pitfall traps were set up with 3 m or 4 m distance in a ‘X’ outline (Figure 2). For each pitfall trap (a 475 mL PET cup, with an upper diameter of 8 cm, lower diameter of 5.5 cm, and 13 cm depth), it was filled by one-third with a mixture of saturated salt water to prevent samples from rotting, and then dishwashing liquid was used to remove the surface tension. The top edge of the pitfall trap was flush with the ground. The center of each sample plot represented results obtained from nine pitfall traps. Sampling was conducted at 10-day intervals during the whole sampling period, i.e., 8 times in 2019 (mid-November to mid-February the following year) and 14 times in 2020 (mid-November to mid-March the following year). After each sampling, traps were emptied and refilled. Pitfall contents were preserved in 75% ethanol, adult spiders and carabids were identified at the species level, and the remaining individuals that could not be identified at the species level were identified at the genus or family level.

### 2.3. Spider and Carabid Community Traits

To analyze the functional diversity of spiders and carabids, we selected three functional traits: body size, dispersal ability, and feeding [36,37]. For spiders, the body size and dispersal ability were classified according to the World Spider Catalog [38] and Bell et al. (2005) [39]. Those with a body size equal to or longer than 5 mm (>=5 mm) were classified as large spiders, while those shorter than 5 mm (<5 mm) were classified as small spiders [40]. All captured spiders were considered predators [41]. Ballooning species were classified as code 1, and no-ballooning spider species as code 0 [42].

For carabids, it was suggested that those with a body size equal to or longer than 15 mm be classified as large carabids and those shorter than 15 mm be classified as small carabids [43]; however, we did not find any carabids with a body size longer than 15 mm. We thus distinguished between a body size >= 7.5 mm (as large carabids) and a body size < 7.5 mm (as small carabids). The mean body size of carabids was measured in at least 20 individuals for each species, and if the sample size was smaller than 20, all individuals were measured. The carabids’ feeding traits were categorized into two feeding preferences: carnivorous and omnivorous [44,45]. All the captured carabids were macropterous species whose wings were longer than the elytrae [46]. The functional traits of spiders and carabids are listed in Table 1.

### 2.4. Data Analysis

We separately calculated the spiders and carabids per pitfall trap, to obtain densities in 2019 (8 rounds) and 2020 (14 rounds), as well as the species richness for the two different habitats, to avoid any influence of missing or broken traps. Here, we only considered adult spiders and carabids in total. All statistical analyses were performed using the RStudio platform [47].
i.In order to analyze the spiders’ and carabids’ density between different months in the rice fallow field and oilseed rape field, the alpha diversity (abundance, richness, diversity), and functional traits (for spiders: body size, dispersal ability; for carabids: feeding trait, body size) during the sampling period, a paired *t*-test or the Kruskal–Wallis test was used. We also calculated community-weighted mean values (CWMs) of spiders’ and carabids’ body sizes between the fallow rice field and oilseed rape field, which were calculated as follows:
(1)$$\mathrm{CWM}=\sum_{i=1}^{S}pixi$$
where CWM is the community-weighted mean value of a given functional trait, *p_i_* is the relative abundance of species *i* (*i* = 1, 2, …, *S*), and *x_i_* is the trait value for the species [48].

ii.We performed a non-metric multidimensional scaling (NMDS) ordination based on the Bray–Curtis distance matrix from the relative abundance of all adult species using the “vegan” package in R [49], to visualize the spatial community dissimilarity of spiders and carabids in fallow rice fields and oilseed rape fields. Subsequently, PERMANOVAs (999 permutations) were used to test for significant difference in species assemblages between the oilseed rape fields and the fallow rice fields.iii.Finally, we analyzed the landscape composition within a radius of 1000 m around the focal patch. We used a redundancy analysis (RDA) to assess the interactions between landscape variables and all ground predators’ composition, and to visualize the responses of significant indicator species in biplots of the RDA. Using the species composition as the dependent variables, landscape composition as the argument variables in the RDA, and stepwise forward selection to find the optimum model with predictive redundancy, we calculated the variance inflation factor (VIF < 5) to firmly exclude multicollinearity between explanatory variables [50]. The best-explained variation was selected in spiders and carabids with 999 Monte Carlo permutations. To avoid visual confusion, we only show the significant indicator species in the redundancy analysis. The other full figures are attached in Appendix A. RDA analyses were carried out using the “vegan” package in R [49].

## 3. Results

During the two sampling periods, we collected a total of 15,554 adult spiders belonging to 34 species. The dominant spider was *Ummeliata insecticeps* Bosenberg et Strand (67.20% of total spider individuals), followed by *Erigone prominens* Bosenberg et Strand (12.40% of total spider individuals). The proportion of large spiders was 9.79%, while small spiders made up 90.21%. The proportion of ballooning spiders was 91.30% and that of no-ballooning spiders was 3.76% (Appendix A). In the case of carabids, in total, 1549 individuals were collected, which were further classified into 12 carabid species. The dominant carabids were *Agonum chalcomus* Bates (42.40% of total carabid individuals), followed by *Pterostichus liodactylus* (Tschitscherine) (31.30% of total carabid individuals). The proportion of predator carabids was 96.00%, and the proportion of omnivore carabids was 4.00% (Appendix A).

### 3.1. Monthly Dynamics of Spider and Carabid Density in FRs and OSRs

Spider density in FRs was higher as compared to OSRs in all sampling months except for November (Figure 3A). Seasonal changes in spider density in FRs were significantly higher than that in OSRs in December (t = −2.87, *p* = 0.01) and February (t = −2.73, *p* = 0.003) (Figure 3A).

Carabid density in OSRs was higher than that in FRs (Figure 3B). Seasonal changes in carabid density were significantly higher in OSRs compared to FRs in January (z = −1.96, *p* = 0.04) and February (z = −2.10, *p* = 0.02) (Figure 3B).

### 3.2. Alpha Diversity and Functional Traits of Spiders and Carabids

There was no significant difference in alpha diversity between FRs and OSRs for carabid assemblages; however, for spider assemblages, species abundance in FRs was significantly higher than that in OSRs (Figure 4A, t = −2.762, *p* = 0.015), and more spiders preferred the FRs to the OSRs. However, we found no effect of habitat types on the community-weighed mean (CWM) values of body size in either spiders or carabids (Figure 4A,B).

Habitats had a significant effect on the individual body sizes of spiders and carabids; the proportion of smaller spiders (body size < 5 mm) in FRs were significantly higher than those in OSRs (z = −3.309, *p* < 0.001). Meanwhile, the proportion of larger carabids (body size >= 7.5 mm) in FRs were significantly higher than those in OSRs (z = −2.551, *p* = 0.012), whereas the OSRs had a negative effect on large carabids (body size >= 7.5 mm). Spiders in the FRs showed a higher dispersal power than in the OSRs (t = −4.490, *p* < 0.001), with more ballooning spiders observed in the fallow rice fields, while predator carabids were more common both in the FR and OSR habitats (z = −0.770, *p* = 0.441).

### 3.3. Composition of Spider and Carabid Assemblages in the FRs and the OSRs

For spiders, the NMDS analysis demonstrated that the spider assemblages were more clustered in the FRs than that in the OSRs, however, without significant differences (PERMANOVA: F_(1,30)_ = 2.077, *p* = 0.060, *R*^2^ = 0.065, Figure 5A), indicating that the spider assemblages had a relatively high homogeneity both in the FR and OSR habitats. For carabids, the community assemblages were significantly affected by the habitats of OSRs and FRs (PERMANOVA: F_(1,29)_ = 2.836, *p* = 0.008, *R*^2^ = 0.089, Figure 5B), with carabids preferring to congregate in the OSRs, and with a greater homogeneity in the OSRs than in the FRs.

### 3.4. Effect of Habitat Types and Landscape Compositions on Spider and Carabid Assemblages

RDA analyses were used to explain the effect of landscape compositions on the spider and carabid assemblages, in which the landscape variables were assessed for a focal central sample field with a radius of 1000 m. The RDA model indicated that the percentage of oilseed rape area positively correlated with several spider species, with the exception being *U. insecticeps* (F_(1,29)_ = 8.380, *p* = 0.001, Figure 6A). The *U. insecticeps* density was positively affected by the proportion of semi-natural habitats (F_(1,29)_ = 2.890, *p* = 0.023, Figure 6A). The individual density of three carabid species (*Bembidion perditum* (Netolitzky)*, Stenolophus kurosai* (Tanaka)*,* and *A. chalcomus*) was positively related to the proportion of OSR. Conversely, *P. liodactylus* and *Agonum japonicum* (Motschulsky) individuals were negatively affected by the proportion of OSR (F_(1,30)_ = 2.430, *p* = 0.001, Figure 6B).

## 4. Discussion

Taken together, the findings of this study reveal the response of spiders and carabids to different planting regimes and surrounding habitats during overwintering. We have found that FRs may support more active spiders and a higher density of spiders (especially for small species), while OSRs better support carabids. Furthermore, the community structure of spiders we observed was affected by both planting regimes and semi-natural habitats, while the community structure of carabids was mainly regulated by planting regimes.

### 4.1. Effect of Habitat Types on Density and Diversity of Predators

According to the seasonal active density of spiders and carabids, we found that spiders preferred the FRs while more carabids preferred the OSRs. The ballooning spider species had a strong dispersal ability, while fallow fields contained more bare land and a lack of rape vegetation to provide shade, which was more conducive to ballooning [51], as our results confirmed. In addition, spider assemblages were less affected by land use management, like mowing or temporary grazing [52,53], than the carabid assemblages. After the rice had been harvested, spiders could recolonize environments through ground activity or through passive aerial dispersion. Another finding was that all the carabids collected in the pitfall traps were larger than the spiders, which possibly indicates that the carabids had a higher need for resources than the spiders, and the OSRs with a higher density of pests may have provided a certain trophic resource. As reported in a previous study, when we compared the woody and grassy strips established in the direct vicinity of the cropland, we found that the OSR strips harbored the highest carabid species richness and activity density [54]. Sivcev et al. (2018) also found that the density of ground beetles in OSRs was 14 times higher than that in winter wheat fields [55], which may reflect a strong influence of habitat types on over-wintering beetle assemblages, and which may indicate that OSRs play a highly beneficial role for carabid species [56]. As such, a rice–rape rotation system may have great potential to enhance the complementary effects between the two ground predators.

### 4.2. Effect of Two Habitat Types on Functional Traits’ Diversity in Spiders and Carabids

Incorporating functional trait diversity could provide a more comprehensive understanding of the relationship between biodiversity and different habitat types [57,58]. Overall, we found that the FRs contained more small-body-size (<5 mm) spiders and ballooning spiders than the OSRs, which was consistent with the previous finding that small and ballooning spiders often colonize crop fields [59]. Overwintering spider assemblages were more strongly affected by vegetation type than the carabid assemblages. There is support for this in the literature, for example, with the finding that Linyphiidae always prefer to overwinter in herbaceous habitats rather than in woody or semi-natural habitats [60]. A potential reasonable explanation for this is that the Linyphiidae can more quickly colonize the fallow fields (especially due to small body sizes and ballooning immigration) after the crop has been harvested. Compared to spiders, we found that the density of carabids was higher in oilseed rape fields than in fallow fields, especially for small carabids (<7.5 mm). Previous research works had shown that larger ground beetles are more migratory and more sensitive to environmental changes [61]. For example, when oilseed rape is growing, the larger carabids have greater dispersal ability to adjacent habitats [24]. Thus, larger predators, no matter whether spiders or carabids, seem to be more sensitive to agricultural practices than smaller predators. In addition, spillover effects are common in ground beetles when oilseed rape is growing, where large carabids in fallow fields may migrate under the influence of land use management operations or from adjacent semi-natural habitats [24,62]. Meanwhile, smaller carabids are often considered better in terms of their adaptability to farmland habitats such as the OSRs, with their more regular and stronger disturbance regimes [63,64]. Besides migration, relationships between resource and habitat complexity can also explain the mechanism at work between resource use and habitat types, such as through the notion that increasing landscape complexity mainly affects carabids positively, and with the finding that epigaeic beetles have higher biological control of aphids in complex landscapes [65].

### 4.3. Effect of Habitat Types and Landscape Compositions on Spider and Carabid Assemblages

Different farmland landscape patterns and their connectivity, scales, and change processes can profoundly affect the diversity, composition structures, and migration dynamics of natural enemies [66,67]. Here, we found that SNHs had a positive effect on sheet-web and small spiders like *U. insecticeps*, with OSRs still able to support ground-hunting and large spiders like Lycosidae during overwintering to some extent. As far as we know, they consume pollen beetles in OSRs [68]. Our study is consistent with most previous studies in showing that SNHs have stable heterogeneous environments, which are conducive to the reproduction of natural enemies [69,70]. After crop harvesting, arthropods in paddy fields will migrate to the semi-natural habitats nearby (such as different grass species) to avoid external stress imposed by different planting regimes [71]. Mestre et al. (2018) found that SNHs harbor a higher diversity and density of overwintering spiders than crop fields [60]. In addition, Drapela et al. (2008) reported a higher diversity of spiders in OSR fields, where the spider species richness in the OSRs was increased with the increasing of OSR habitats [72,73].

For carabids, the OSRs can also provide larger and bushy vegetated areas, which serve as ideal hunting grounds, containing plenty of eggs, larvae, and pupae of chrysomelids and dipterans [74,75]. However, we found two dominant carabids, *Pterostichus melanarius* (Illiger) and *P. madidus* (Fabricius), were negatively affected by the OSRs. Although we made some efforts in evaluating the alpha diversity and functional diversity to disentangle the differences between the two dominant carabids, maybe we were missing some other vital distinctions. Carabids of *P. melanarius* and *P. madidus* are morphologically similar and we found they had similar feeding traits and responses the OSRs. They were negatively affected by the OSRs, likely due to niche differentiation, where different species can find their places in the same landscape habitat, avoiding direct competition [76,77,78].

## 5. Conclusions

We found variable responses of spiders and carabid beetles to different overwinter planting regimes and landscape compositions, which could be grounded in ecological requirements and functional traits. Specifically, our study indicated that rape field increased the activity and diversity of carabid beetles, while fallow field increased the activity and diversity of spiders, especially for smaller and ballooning spiders. Winter oilseed rape field played a prominent role in conserving the diversity of carabid beetles, while winter fallow field was beneficial to the diversity of spiders. Therefore, we recommend reserving some fallow fields when rapeseed is planted in winter, which can effectively improve the diversity of predatory natural enemies in farmland. These findings may lead to improved methods of enhancing sustainable rice ecosystem services in the coming years.

## Figures and Tables

**Figure 1 insects-14-00951-f001:**
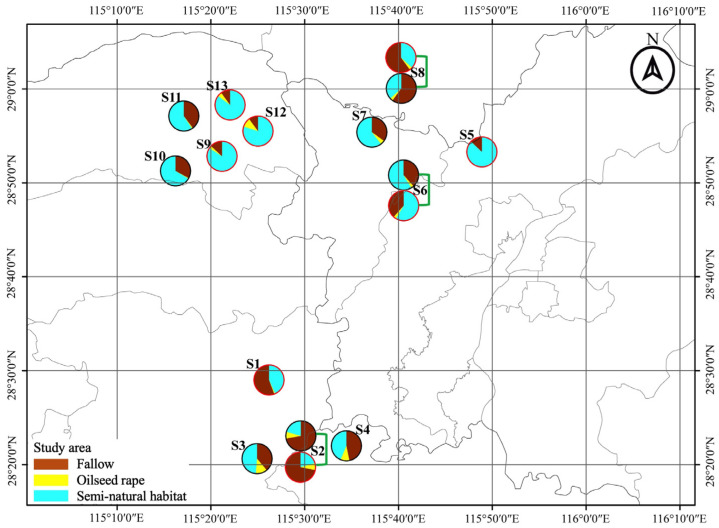
Locations of fallow rice (FR) and oilseed rape (OSR) sampling fields in the region of Jiangxi Province, China during overwinters of 2019/2020 and 2020/2021. Pies show the composition of the landscape at a 1.0 km radius around focal fields.

**Figure 2 insects-14-00951-f002:**
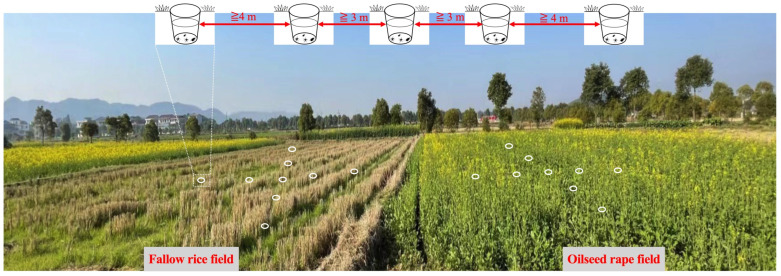
Sketch distribution of X-shaped arrangement of nine pitfall traps set up in adjacent FR and OSR.

**Figure 3 insects-14-00951-f003:**
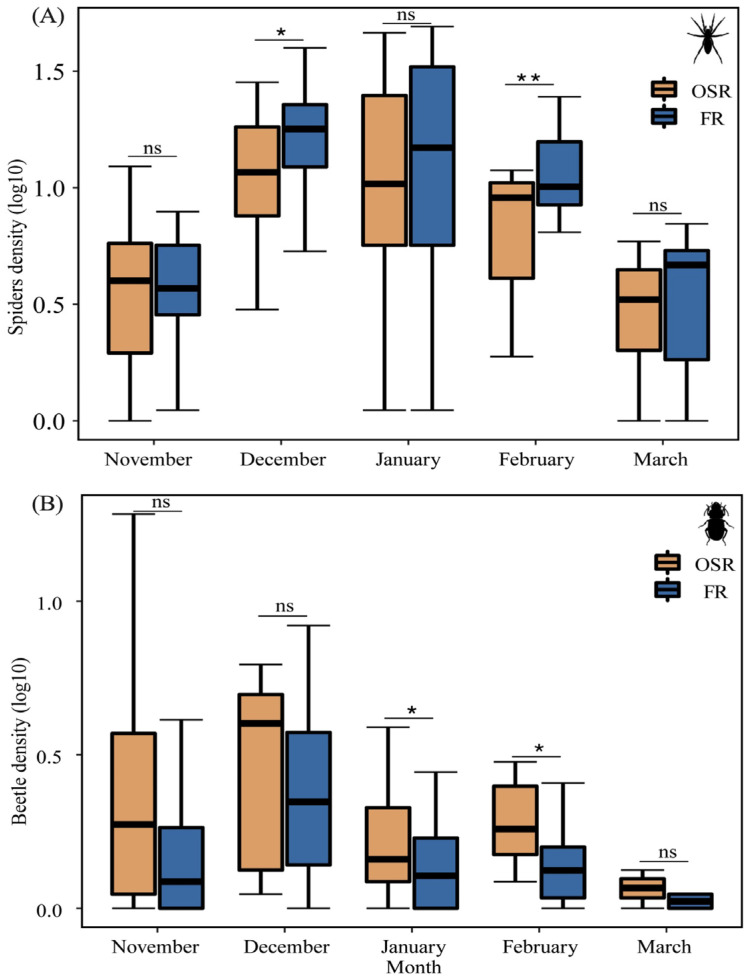
Activity density (mean, log10-transformed) of spiders (**A**) and carabids (**B**) in winter FRs (*n* = 8) and OSRs (*n* = 8). * *p* < 0.05, ** *p* < 0.01, ns is not significant.

**Figure 4 insects-14-00951-f004:**
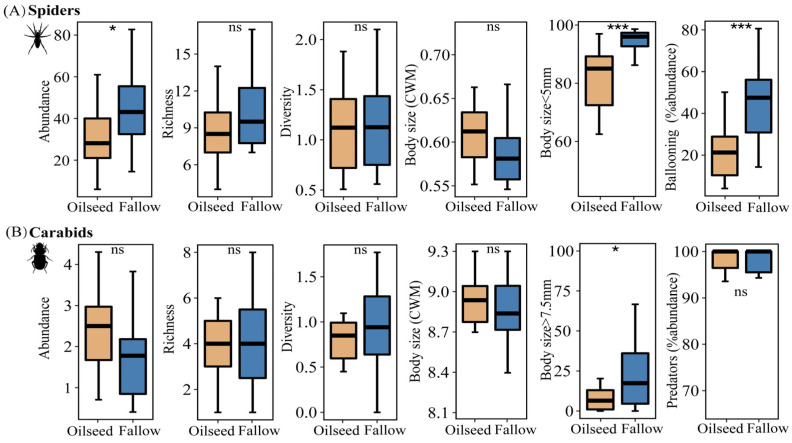
Comparison of the alpha diversity and functional traits of spiders (**A**) and carabids (**B**) between fallow rice fields and oilseeds rape fields. In the boxplots, the box represents the interquartile range (25–75%) and the band inside is the median. Differences were tested with a paired *t*-test or the Kruskal–Wallis test. ns is not significant, * *p* < 0.05, *** *p* < 0.001.

**Figure 5 insects-14-00951-f005:**
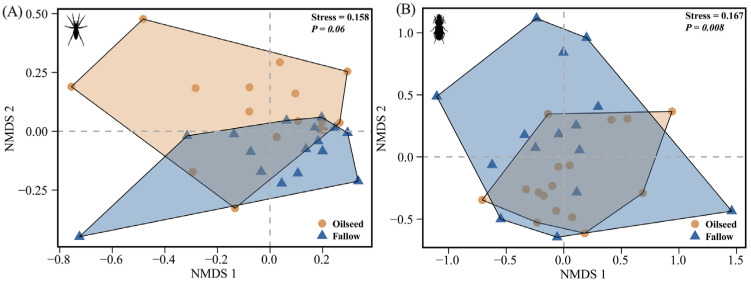
Non-metric multidimensional scaling analysis (NMDS) for (**A**) spiders based on 34 species and 15,554 individuals, and (**B**) carabids based on 12 species and 1549 individuals. All species combined across two habitat types (*n* = 16 plots, 2 dimensions, Bray–Curtis distance).

**Figure 6 insects-14-00951-f006:**
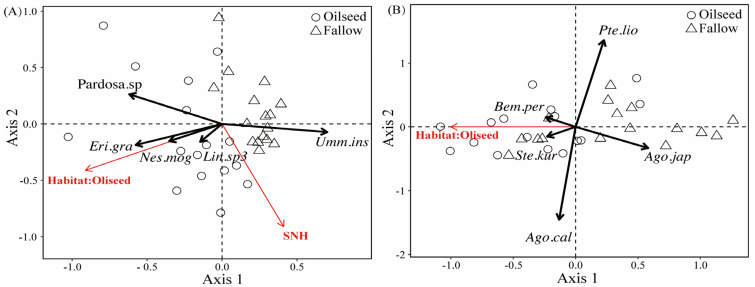
RDA ordination diagrams of spiders (**A**) and carabids (**B**) in FR and OSR, marking significant landscape variables and significant indicator species (crosses) along the first and second RDA plural axes. SNH, semi-natural habitats. (1) *Eri.gra*, *Erigonidium graminicolum*; (2) *Nes.mog*, *Nesticella mogera*; (3) *Lin.sp3*, *Linyphiidae.sp3*.; (4) *Bem.per*, *Bembidion perditum*; (5) *Ste.kur*, *Stenolophus kurosai*; (6) *Ago.cal*, *Agonum chalcomus*; (7) *Ago.jap*, *Agonum japonicum*; (8) *Pte.lio*, *Pterostichus liodactylus*.

**Table 1 insects-14-00951-t001:** Functional traits of spiders and carabids.

Functional Traits	Trait Variables	
	Spiders	Carabids
Body size	Sizes: 1–5, 5–10 mm	Sizes: < 7.5 mm, > 7.5 mm
Dispersal ability	Ballooning, no ballooning	Wide wing type
Feeding trait	Predator	Carnivorous, omnivorous

## Data Availability

All data are contained in the manuscript.

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
