# Peer review of "Response of Spider and Epigaeic Beetle Assemblages to Overwinter Planting Regimes and Surrounding Landscape Compositions"

_insects, 2023, doi:10.3390/insects14120951_

Round 1
Reviewer 1 Report
Comments and Suggestions for Authors
Please correct the following mistakes.
Line 166 elitrae - correctly elytrae
Line 259 carbides - correctly carabids
Line 276 carbides - correctly carabids
Line 278 first and second RDA axe - correctly axis or plural axes
Line 309 4.2. Effect of two habitat types on functional traits diversity of spider and carabid - use the plural form spiders and carabids
Line 319 ...we found carabids density - correctly carabid density or density of carabids
Line 375 carbides - correctly carabids
Line 377 along the first and second RDA axe - correctly plural is axes or singular axis
Reviewer 2 Report
Comments and Suggestions for Authors
Dear authors,
After reviewing the attached manuscript, I can say that it is a complex and interesting work that is always up-to-date and can significantly contribute to the knowledge of the structure and density of the arthropod community in agricultural ecosystems. I believe that the manuscript is ready for acceptance and publication in the journal Insects, with minor changes, mostly in the literature section.
One of the more significant changes I expect you to make would be a clearer clarification of the pies in Figure 1. The color markings and proportions are completely unclear. Furthermore, what is site 1 and what is site 13, for example, GPRS coordinates are also missing. I expect significant progress in this segment of work.
Indicate by which key the ground beetles were determined!
Latin names of species should be written in the form of binomial nomenclature for naming organisms, in such a way that the author and year are added from the species name. When the first time the species is mentioned in the paper full and later the abbreviated form should be used, I believe that this principle should be used and rearrange in the entire paper because the names of the organisms you determined are not presented in this way.
Line 62: add “ground beetles”
Line 302: check the literature and the way of referencing in the text.
Line 303: check the literature because it is completely inconsistent and incorrect.
Line 342: check the literature because it is completely inconsistent and incorrect.
Line 344: check the literature and the way of referencing in the text.
Before consulting the literature, please look at the following works that deal with similar or the same issue that you describe in the introduction of the work, in order to improve and strengthen it with additional research.
Literature:
Benítez, Hugo A. ; Lemić, Darija ; Püschel, Thomas A. ; Virić Gašparić, Helena ; Kos, Tomislav ; Barić, Božena ; Bažok, Renata ; Pajač Živković, Ivana: Fluctuating asymmetry indicates levels of disturbance between agricultural productions: An example in Croatian population of Pterostichus melas melas (Coleptera: Carabidae) // Zoologischer anzeiger, 276 (2018), str. 42-49. DOI: 10.1016/j.jcz.2018.07.003
I wish you luck in your work and a successful publication in the chosen journal.
With respect.
Reviewer 3 Report
Comments and Suggestions for Authors
Dear Authors, I really appreciate papers dealing with biodiversity studies. I know how much work and how many multidisciplinary competencies they need. You treated the issue in the right way and with the right statistical tecniques. I have just put some comments and advics in the pdf version with the "introduce comments" function. Please put attention in reporting the name of the authors when you cite a species for the first time.
